# Learning Invariant Representations with Missing Data

**Mark Goldstein**                                          GOLDSTEIN@NYU.EDU
**Aahlad Puli**                                                   AAHLAD@NYU.EDU
**Rajesh Ranganath**                                    RAJESHR@CIMS.NYU.EDU
*New York University*

**Jörn-Henrik Jacobsen**                          JHJACOBSEN@APPLE.COM
**Olina Chau**                                                  OLINA@APPLE.COM
**Adriel Saporta**                                        ASAPORTA@APPLE.COM
**Andrew C. Miller**                                   ACMILLER@APPLE.COM
*Apple*

**Editors:** Bernhard Schölkopf, Caroline Uhler and Kun Zhang

## Abstract

Spurious correlations allow flexible models to predict well during training but poorly on related test populations. Recent work has shown that models that satisfy particular independencies involving correlation-inducing *nuisance* variables have guarantees on their test performance. Enforcing such independencies requires nuisances to be observed during training. However, nuisances, such as demographics or image background labels, are often missing. Enforcing independence on just the observed data does not imply independence on the entire population. Here we derive MMD estimators used for invariance objectives under missing nuisances. On simulations and clinical data, optimizing through these estimates achieves test performance similar to using estimators that make use of the full data.

**Keywords:** invariant representations, missing data, doubly robust estimator, MMD

## 1. Introduction

Spurious correlations allow models that predict well on training data to have worse than chance performance on related populations at test time (Geirhos et al., 2020; Puli et al., 2021; Veitch et al., 2021; Makar et al., 2021; Gulrajani and Lopez-Paz, 2020; Sagawa et al., 2020). For example, diabetes is associated with high body mass index (BMI) in the United States. However, in India and Taiwan, diabetes also frequently co-occurs with low and average BMI (WHO, 2004). Due to their shifting relationship with the label, nuisance variables (e.g., BMI) can cause models to exploit correlations in training data, leading them to generalize poorly on test sets of interest.

*Invariant prediction* methods are designed to improve performance on a range of test distributions when training data exhibits spurious correlations (Peters et al., 2016; Arjovsky et al., 2019). We focus on methods that enforce independencies between the model and nuisance given some assumed causal structure (Makar et al., 2021; Veitch et al., 2021; Puli et al., 2021). These methods require the nuisance to be specified explicitly and observed. However, in large health datasets, nuisances are often missing. For example, not all people who report diabetes status report other correlated conditions (e.g., hypertension, depression) or demographics (e.g., gender).

To improve generalization on a range of test distributions, it is necessary to handle missingness appropriately. However, extending invariant prediction methods to handle missing data is not straightforward. This difficulty stems from the invariant method's optimization objective, which

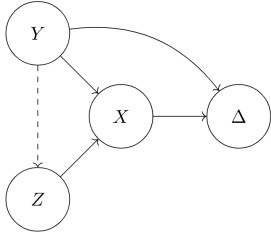

**Figure 1:** Generative process we consider in this work. The $Y \to Z$ edge is dashed to emphasize that the $Z|Y$ may change at test time. $\Delta$ determines missingness of $Z$ and satisfies $Z \perp\!\!\!\perp \Delta | (X, Y)$.

usually includes a measure of dependence — e.g., Maximum Mean Discrepancy (MMD) or Mutual Information (MI) — that requires a sample from the fully-observed data to estimate consistently.

We propose MMD estimators for measuring nuisance-model dependence under missingness. First, we show that enforcing independence on only the nuisance-observed data does not imply independence on the full data, and vice versa. Next, we derive three estimators, including one that is *doubly-robust*: it is consistent when either the nuisance or missingness can be consistently modeled (Bang and Robins, 2005). Using simulations, a semi-simulation using textured MNIST and clinical data from MIMIC, we show that the estimators perform close to ground-truth estimation with no missingness and that they improve test accuracy relative to MMD computation using only the data with nuisances observed.

## 2. Notation and background

**Notation.** Let $X$ denote features. Let $Y$ be a label such as disease status. Let $Z$ denote the nuisance, e.g., another disease correlated with $Y$, demographics, or image backgrounds. Denote the nuisance missingness indicator as $\Delta$. Instead of $(X, Y, Z)$, we observe $(X, Y, \Delta, \tilde{Z} = \Delta Z)$, where $\tilde{Z} = Z$ when $\Delta = 1$ and $Z$ is unobserved otherwise. We write functions as $f_X = f(X)$ to avoid excess parentheses. Let $h_X = h(X)$ denote a model to predict $Y$. When conditioning on events involving both $X, Y$ we use $W = (X, Y)$ and $w = (x, y)$ to lighten notation so that $\mathbb{E}[\Delta | X = x, Y = y] = \mathbb{E}[\Delta | W = w]$ and $f(X, Y) = f_{XY} = f_W$. Let $\mathcal{B}(p)$ denote Bernoulli($p$).

**Assumptions and scope.** The estimators require ignorability, $Z \perp\!\!\!\perp \Delta | W$ (Hernan and Robins, 2021). Some distributions that satisfy this are shown in Figure 1. We require positivity $0 < \epsilon \leq P(\Delta = 1 | W)$ to observe $Z$ appropriately. While we focus on the graph in Figure 1 with binary $Z$, the presented method for handling missingness in invariance objectives can extend to continuous $Z$ (Appendix H) and to other data generating processes so long as (1) the distribution shift assumptions of the underlying invariance method are met and (2) ignorability $Z \perp\!\!\!\perp \Delta | W$ is satisfied.

**Modeling under spurious correlations.** Nuisance-based prediction arises in training data when $Z$ is predictive of $Y$ and associated with $X$, causing models to use information about $Z$ in $X$ to predict $Y$. This may not be a problem in all scenarios, but it is when the test distribution is expected to have a different $(Y, Z)$ relationship from the training distribution, as this may imply $P_{train}(Y | X) \neq P_{test}(Y | X)$ and a model built on training data may not perform well on test data. Most approaches to this problem start off by narrowing down the set of possible test distributions and their relationship to the training distribution. In this work, we focus on a family studied in

Makar et al. (2021); Veitch et al. (2021); Puli et al. (2021). The distributions are indexed by $D$ and vary only in the factor $P(Z \mid Y)$:

$$\mathcal{F} = \{P_D(X, Y, Z) = P(Y)P_D(Z|Y)P(X|Y, Z)\}_D. \tag{1}$$

When $P_{test}(Z|Y) \neq P_{train}(Z|Y)$, in general $P_{train}(Y \mid X) \neq P_{test}(Y|X)$ and a model built on $P_{train}$ can generalize poorly. For example consider, for $a \in \mathbb{R}$,

$$Y \sim \mathcal{B}(0.5), \quad \mu_Z = a(2Y - 1), \quad Z \sim \mathcal{N}(\mu_Z, 1).$$

When $P_{train}$ uses $a = 0.5$ and $P_{test}$ uses $a = -0.9$, Puli et al. (2021) show an example of an $X|Y, Z$ distribution where an ERM model with $80\%$ training accuracy only achieves $40\%$ on the test set, due to the changing relationship between $Y, Z$.

When it is possible to anticipate and observe nuisances during training, enforcing certain independence constraints (Makar et al., 2021; Veitch et al., 2021; Puli et al., 2021) helps guarantee performance regardless of the nuisance-label relationship. For example, for this choice of $\mathcal{F}$ and model $h$, maximum likelihood estimation for $Y|h_X$ while enforcing the constraint $h_X \perp\!\!\!\perp Z \mid Y$ implies equal performance on all $P_D \in \mathcal{F}$, and better than chance performance (Appendix A).

**Measuring Dependence.** To enforce independencies it is necessary to measure dependence and to minimize this measure. One way to measure dependence is to measure distance between a joint distribution and the product of its marginals. The general Integral Probability Metrics (IPMs) class defines a metric on distributions. A special case, the kernel-based MMD (Gretton et al., 2012), has a closed form. Let $X_1 \sim P, X_2 \sim Q$. Let $X'_j$ be an independent sample identically distributed as $X_j$. For kernel $k$,

$$\text{MMD}_k(P, Q) = \mathbb{E}[k(X_1, X'_1)] + \mathbb{E}[k(X_2, X'_2)] - 2\,\mathbb{E}[k(X_1, X_2)]. \tag{2}$$

The MMD is $0$ if and only if $P =_d Q$ under certain mild conditions on $P, Q$ and the kernel $k$ (Gretton et al., 2012). Therefore, computing the MMD on a joint distribution of two variables and the product of its marginals is a measure dependence, which also coincidences with the Hilbert-Schmidt Independence Criterion (HSIC) (Gretton et al., 2005; Szabó and Sriperumbudur, 2017).

**Estimation under missingness.** The problem that this work tackles is enforcing conditional independencies such as $h_X \perp\!\!\!\perp Z|Y$ to improve generalization in families like $\mathcal{F}$ in Equation (1) even when $Z$ is subject to missingness. Methodology from causal inference solves a related problem: estimating $\mathbb{E}[Z]$ when $Z$ is subject to missingness. In this work, we extend this methodology from estimates of $\mathbb{E}[Z]$ to estimates of objectives that enforce independencies involving $Z$. Here we review estimation of $\mathbb{E}[Z]$ under missingness. Two parts of the data-generating distribution can help. Letting $W = (X, Y)$, define:

$$
\begin{aligned}
G_W &\triangleq \mathbb{E}[\Delta \mid X, Y] &&\text{(missingness process)} \\
m_W &\triangleq \mathbb{E}[Z \mid X, Y] &&\text{(conditional expectation)}
\end{aligned}
$$

We review estimators of $\mathbb{E}[Z]$ that use $G_W$ (Horvitz and Thompson, 1952; Binder, 1983; Robins et al., 1994) or $m_W$ (Rubin, 1976; Schafer, 1997) in Appendix E. The *doubly-robust* (DR) estimator

(Robins and Rotnitzky, 2001; Bang and Robins, 2005; Kang and Schafer, 2007) combines both by noting the following equality:

$$\mathbb{E}[Z] = \mathbb{E}\left[\frac{\Delta \tilde{Z}}{G_W} - \frac{\Delta - G_W}{G_W}m_W\right]. \tag{3}$$

Crucially, the right side of Equation (3) does not requires samples of $Z$ when $\Delta = 0$. When $G_W$ or $m_W$ are replaced with estimates $\hat{G}_W, \hat{m}_W$, the equality still holds provided that for all $w$, *either* $\hat{G}_w = G_w$ *or* $\hat{m}_w = m_w$ (Appendix F). Moreover, Monte Carlo estimates of the right side of Equation (3) are consistent for $\mathbb{E}[Z]$ when *either* $\hat{G}_W$ consistently estimates $G_W$ or $\hat{m}_W$ consistently estimates $m_W$ (Robins and Rotnitzky, 2001). This is useful because in practice neither of $G_W$ nor $m_W$ are known and both must be estimated.

## 3. Invariant representations with missing data

For the graph in Figure 1 and binary $Z$, Veitch et al. (2021) enforce $h_X \perp\!\!\!\perp Z|Y$ for the predictive model by maximizing likelihood while minimizing the MMD:

$$\max_h \quad \log p(y|h_X) - \lambda \cdot \sum_{y \in \{0,1\}} \text{MMD}_k\left(p(h_X|Z=1, Y=y), p(h_X|Z=0, Y=y)\right). \tag{4}$$

The first term is the usual maximum-likelihood objective for predicting $y$ with model $h_X$. The second term, because $Z$ is binary, enforces $h_X \perp\!\!\!\perp Z|Y$ at optimum. First, we motivate the independence constraint for the choice of assumed family $\mathcal{F}$ in Equation (1). We then demonstrate what can go wrong when enforcing this MMD penalty only on samples where $Z$ is observed. We then derive estimators of the full-data MMD under missingness.

### 3.1. Conditional Independence implies equal performance on the anti-causal family

There are at least two distinct usages of the word *invariance* in the literature. One refers to independence (e.g. of a model to a nuisance or environment variable). The other refers to *invariant risk*, i.e., the risk is the same for all test distributions in some family (Arjovsky et al., 2019; Krueger et al., 2021). In some families and for some independence constraints, these can coincide.

**Proposition 1** *Suppose model $h_X$ satisfies $h_X \perp\!\!\!\perp Z|Y$ on any $P_D \in \mathcal{F}$. Then for all $P_{D'} \in \mathcal{F}$, $\mathbb{E}_{P_D}[\log p_D(Y|h_X)] = \mathbb{E}_{P_{D'}}[\log p_D(Y|h_X)]$.*

This has been shown in Veitch et al. (2021); Puli et al. (2021) but we provide a self-contained proof in Appendix A. This result means that estimates of held-out performance from the training data (one member of $\mathcal{F}$) will represent test performance on other member of $\mathcal{F}$ under $h_X \perp\!\!\!\perp Z|Y$.

### 3.2. Failures of restricting to observed data

Under missingness, we observe $(X, Y, \Delta, \tilde{Z})$ instead of $(X, Y, Z)$ where $\Delta = 1$ means $\tilde{Z} = Z$ and when $\Delta = 0$, $Z$ is unobserved. When $Z$ is subject to missingness, we cannot directly compute empirical estimates of the MMD. What happens when we compute the MMD only on samples with $Z$ observed (i.e. conditioning on $\Delta = 1$)? Let us refer to this as the *observed-only* MMD. Restricting computation to data with non-missing $Z$ enforces $h_X \perp\!\!\!\perp Z|Y = y, \Delta = 1$ instead of $h_X \perp\!\!\!\perp Z|Y = y$. We show that these conditions do not imply each other in general.

**Proposition 2** *There exist distributions on $(X, Y, \Delta, Z)$ such that*

$$\exists h_X^\star \quad \text{s.t.} \quad h_X^\star \perp\!\!\!\perp Z | Y = y, \quad \text{but} \quad h_X^\star \not\perp\!\!\!\perp Z | Y = y, \Delta = 1$$

*and there exist distributions on $(X, Y, \Delta, Z)$ such that*

$$\exists h_X^\star \quad \text{s.t.} \quad h_X^\star \perp\!\!\!\perp Z | Y = y, \Delta = 1 \quad \text{but} \quad h_X^\star \not\perp\!\!\!\perp Z | Y = y$$

The proof is in Appendix D. This existence implies:

1. Optimizing the observed-only MMD can discard a solution to the full-data MMD

2. Using the observed-only MMD may lead one to believe a model is invariant when it is not.

To keep the generalization guarantees one must enforce independence on the *full data distribution*.

### 3.3. MMD estimation under missingness

We present estimators of the full-data *unconditional* MMD under missing $Z$, which enforces $h_X \perp\!\!\!\perp Z$ (unconditional on $Y$). Everything that follows can be conditioned on $Y = y$ to enforce $h_X \perp\!\!\!\perp Z | Y = y$ simply by restricting samples used to estimate the expectations to those with $Y = y$. For a kernel $k$ let $k_{XX'} \triangleq k(h_X, h_{X'})$. The MMD can be written as:

$$\text{MMD}\big(p(h_X | Z = 1), p(h_X | Z = 0)\big) = \mathop{\mathbb{E}}_{\substack{X | Z = 1 \\ X' | Z' = 1}} k_{XX'} + \mathop{\mathbb{E}}_{\substack{X | Z = 0 \\ X' | Z' = 0}} k_{XX'} - 2 \mathop{\mathbb{E}}_{\substack{X | Z = 1 \\ X' | Z' = 0}} k_{XX'}. \quad (5)$$

Estimation is challenging due to missingness in the conditioning set. For $b \in \{0, 1\}$, let $N(b, b') = P(Z = b)P(Z' = b')$ and let $Z_1 \triangleq Z$ and $Z_0 \triangleq 1 - Z$. The dependence on $Z$ can be re-written:

$$\mathop{\mathbb{E}}_{\substack{X | Z = b \\ X' | Z' = b'}} k_{XX'} = \frac{1}{N(b, b')} \mathbb{E}\Big[k_{XX'} \cdot Z_b \cdot Z'_{b'}\Big]. \quad (6)$$

Under no missingness, each expectation could be estimated with Monte Carlo. We now develop three MMD estimators. We propose simple $G_W$-based and $m_W$-based estimators in Equations (7) and (8) and then a doubly-robust estimator that combines them in Equation (9).

**Proposition 3** *($G_W$-based re-weighted estimator) Assume positivity, ignorability, and, $\forall w$, $G_w = \mathbb{E}[\Delta | W = w]$. Then,*

$$\mathop{\mathbb{E}}_{\substack{X | Z = b \\ X' | Z' = b'}} \Big[k_{XX'}\Big] = \frac{1}{N(b, b')} \mathbb{E}\Big[\frac{\Delta \Delta' \tilde{Z}_b \tilde{Z}'_{b'}}{G_{WW'}} k_{XX'}\Big]. \quad (7)$$

**Proposition 4** *($m_W$-based regression estimator) Assume ignorability, and, $\forall w$, $m_w = \mathbb{E}[Z | W = w]$. Then,*

$$\mathop{\mathbb{E}}_{\substack{X | Z = b \\ X' | Z' = b'}} \Big[k_{XX'}\Big] = \frac{1}{N(b, b')} \mathbb{E}\Big[m_{Wb} \cdot m_{W'b'} \cdot k_{XX'}\Big]. \quad (8)$$

Let $m_{W1} \triangleq m_W$, $m_{W0} \triangleq 1 - m_W$, and $G_{WW'} \triangleq G_W G_{W'}$.

**Proposition 5** (DR *estimator). Assume positivity, ignorability, and $\forall w$, either $G_w = \mathbb{E}[\Delta | W = w]$ or $m_w = \mathbb{E}[Z | W = w]$. Then,*

$$\underset{\substack{X|Z=b \\ X'|Z'=b'}}{\mathbb{E}} \left[ k_{XX'} \right] = \frac{1}{N(b,b')} \mathbb{E} \left[ \left( \frac{\Delta\Delta' \tilde{Z}_b \tilde{Z}'_{b'}}{G_{WW'}} - \frac{\Delta\Delta' - G_{WW'}}{G_{WW'}} \cdot m_{Wb} \cdot m_{W'b'} \right) k_{XX'} \right]. \quad (9)$$

The proof is in Appendix G. We can use any of Equations (7) to (9) to estimate the terms in eq. (5). Each of Equations (7) to (9) is a ratio of two expectations: the normalization constant $N(b, b')$ depends on $\mathbb{E}[Z]$ and must itself be estimated under missingness (e.g., with Equation (3)). The ratio of consistent estimates of these quantities is consistent by Weak Law of Large Numbers and Slutsky's theorem. We discuss estimation in practice, trade-offs among the three estimators, and variance in Appendix B. We review recent related work in Section 5.

## 4. Experiments

We compare accuracy and MMD minimization using different estimators: NONE (MLE only, no MMD), FULL (MLE and MMD using data with $Z$ fully-observed, which is what could be used as an objective under no missingness), OBS (MLE and observed-only MMD), DR (MLE and DR estimator, called DR+ when using true $G_W$), IP (MLE and re-weighted estimator, called IP+ when using true $G_W$), and REG (MLE and regression estimator).

We first compare these algorithms in a simulation study. We then use textured MNIST to show the utility of the proposed estimators on high-dimensional data. In quantitative tables, we show mean $\pm$ standard deviation over three seeds. We then predict hospital length of stay in the MIMIC dataset, and compare performance when demographic nuisances are subject to missingness. For the $Y|X$ predictive loss, we use negative Bernoulli log likelihood with logit equal to model output $h_X$.

**Comparing MMDs.** In all tables, the training set MMD to evaluate each method is computed using the ground-truth full-data MMD estimation method (see eq. (6)) to show the actual value of MMD achieved, regardless of optimization method. This is also what the FULL method directly optimizes. True $Z$'s are available in both simulated and real data as missingness is simulated. However, each model trains and validates the $\log p(Y|X) + \text{MMD}$ loss using its own estimation method for MMD.

### 4.1. Experiment 1: Simulation.

We set up strong $(Y, Z)$ correlation. With $\overline{Y} = 1 - Y$, the training and validation sets are drawn:

$$Y \sim \mathcal{B}(0.5), \quad Z \sim \mathcal{B}(.9Y + .1\overline{Y}), \quad X \sim [\mathcal{N}(Y - Z, \sigma_X^2), \mathcal{N}(Y + Z, \sigma_X^2)]. \quad (10)$$

The test set has the opposite relationship $Z \sim \mathcal{B}(.1Y + .9\overline{Y})$. Here $h_X^\star = (X_1 + X_2)/2$ predicts $Y$ with smallest MSE among representations satisfying independence. We construct $\Delta$ to show the failure of computing MMD on the observed-only subset. For this, we use $\hat{Z} \triangleq -(X_1 - X_2)/2$, which is correlated with $Z$. We draw $\Delta$ conditional on $h_X^\star$ and $\hat{Z}$ (both are functions of $X$):

$$Q = \mathbb{1}\left[h_X^\star > 0.6\right] \cdot \mathbb{1}\left[\hat{Z} < 0.6\right], \quad \Delta \sim \mathcal{B}(Q + 0.2\overline{Q}).$$

**Table 1:** Simulation. $\lambda = 1$. NONE has highest MMD and lowest test accuracy. OBS improves over this. The DR and REG methods are able to bring the MMD close to $0.0$ and attain best test accuracy.

|  | NONE | OBS | FULL | DR | DR+ | REG |
|---|---|---|---|---|---|---|
| TR MMD | $0.21 \pm 0.04$ | $0.05 \pm 0.04$ | $0.00 \pm 0.01$ | $0.00 \pm 0.01$ | $0.00 \pm 0.01$ | $0.01 \pm 0.00$ |
| TR ACC | $0.89 \pm 0.00$ | $0.87 \pm 0.00$ | $0.86 \pm 0.01$ | $0.85 \pm 0.01$ | $0.84 \pm 0.02$ | $0.86 \pm 0.00$ |
| TE ACC | $0.67 \pm 0.02$ | $0.77 \pm 0.02$ | $0.80 \pm 0.01$ | $0.81 \pm 0.02$ | $0.81 \pm 0.01$ | $0.79 \pm 0.02$ |

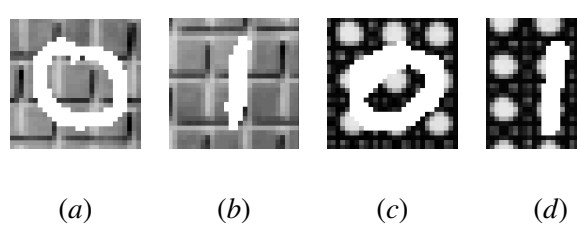

**Figure 2:** Textured MNIST with digits 0,1 on two textures from the Brodatz dataset.

    ($a$)         ($b$)         ($c$)         ($d$)

This example construction leads to $h_X^\star \perp\!\!\!\perp Z|Y$ but $h_X^\star \not\perp\!\!\!\perp Z|Y, \Delta = 1$. For $h, G_W$ and $m_W$ we use small feed-forward neural networks. See the repository[1] for more details.

**Results.** In Table 1, the DR estimators achieve indistinguishable performance to the full-data MMD, both in MMD and accuracy, and better than NONE and OBS. We include more results in Appendix C.

### 4.2. Experiment 2: Textured MNIST.

Following Goodfellow et al. (2013)[2], we correlate MNIST digits 0 and 1 with two textures from the Brodatz dataset (Figure 2). This is an example of invariance to image backgrounds when not all background labels are available. We follow a similar setup to colored MNIST (Arjovsky et al., 2019): because $Y|X$ is essentially deterministic, even strong spurious correlations may be ignored by a model on MNIST. To push $Y|X$ closer to what may be expected in noisier real data, we flip the label with 25% chance. Letting $X$ only predict $Y$ with 75% chance means the model will use texture instead. The missingness is based on the average pixel intensity of $X$ and its class. Let $\mu_X$ be the mean pixel value of a 28x28 MNIST image. We set

$$Q = \mathbb{1}\left[Y = 1\right] \cdot \mathbb{1}\left[\mu_X < 0.3\right], \quad \Delta \sim \mathcal{B}(Q + .2\overline{Q}).$$

The choice of $Q$ is correlated with $Z$ through whether the image is light or dark grey. Similar to proposition 2 and experiment 1, this means subsetting on $\Delta = 1$ does not imply independence on the full population and may throw away solutions that do. For $h, G_W$ and $m_W$ we use small convolutional networks. We include more details in the repository.

**Results.** In Table 2, NONE and OBS perform poorly on test. In contrast, the DR estimators — including the one with a learned $G_W, m_W$ — achieve close to FULL's performance.

---

1. https://github.com/marikgoldstein/missing-mmd
2. We adapt this repository (linked) to construct textured MNIST and will make our code available.

**Table 2:** MNIST $\lambda = 1$. DR and REG estimators achieve close to full performance as measured by full MMD= 0 and high test accuracy. NONE and OBS perform poorly on test. OBS is notably high variance.

|  | NONE | OBS | FULL | DR | DR+ | REG |
|---|---|---|---|---|---|---|
| TR MMD | $2.05 \pm 0.18$ | $0.02 \pm 0.04$ | $0.00 \pm 0.01$ | $0.00 \pm 0.01$ | $0.00 \pm 0.01$ | $0.00 \pm 0.01$ |
| TR ACC | $0.90 \pm 0.01$ | $0.74 \pm 0.03$ | $0.76 \pm 0.01$ | $0.77 \pm 0.0$ | $0.76 \pm 0.01$ | $0.76 \pm 0.01$ |
| TE ACC | $0.13 \pm 0.01$ | $0.63 \pm 0.17$ | $0.74 \pm 0.01$ | $0.72 \pm 0.04$ | $0.73 \pm 0.01$ | $0.73 \pm 0.01$ |

**Table 3:** MIMIC $\lambda = 1$. REG estimator matches FULL's performance and improves upon OBS while DR does not, due to high objective variance during training (not shown in table).

|  | NONE | OBS | FULL | DR | REG |
|---|---|---|---|---|---|
| TR MMD | $0.017 \pm 0.02$ | $0.002 \pm 0.01$ | $0.00 \pm 0.00$ | $0.009 \pm 0.01$ | $0.00 \pm 0.00$ |
| TR ACC | $0.71 \pm 0.02$ | $0.68 \pm 0.01$ | $0.70 \pm 0.01$ | $0.70 \pm 0.01$ | $0.71 \pm 0.00$ |
| TE ACC | $0.64 \pm 0.00$ | $0.64 \pm 0.00$ | $0.66 \pm 0.00$ | $0.62 \pm 0.00$ | $0.66 \pm 0.01$ |

### 4.3. Experiment 3: Predicting length of stay in the ICU

We predict length of stay in the intensive care unit (ICU) in MIMIC (Johnson et al., 2016)[3] using demographics and first day labs/vitals among patients that stay at least one day. The prediction task is whether the stay is more than 2.5 days. To demonstrate that spurious correlations cause issues at deployment, we choose $Z = 1$ to indicate the patient is recorded as white. While race may be correlated with health outcomes (e.g., due to unobserved socioeconomic factors (Obermeyer et al., 2019)), it may not always be appropriate for a model to use this information (Chen et al., 2018). The test set represents a new population with different outcome-demographic structure: we split the data so that the training/validation set has mostly samples with $Y \neq Z$ while the test set has mostly samples with $Y = Z$. We set non-male patients to have $Z$ observed with probability 0.2. We include more details in the repository.

**Results.** In this real data setting with strong $(Y, Z)$ correlation, the full-data MMD estimator reported in the table for all methods may have high variance. We focus on the attained accuracies. The REG estimator matches the ground-truth FULL estimator and performs better than OBS and DR. This is not unexpected, since it is possible for the REG estimator to be lower variance than DR when the true $G_W$ is small or $G_W$ is not modeled well (Davidian, 2005), especially under strong $(Y, Z)$ correlation (Appendix B).

## 5. Related work

We focus on recent work in fairness and invariant prediction on missing group/environment labels. Motivated by fairness, Wang et al. (2020) study a related problem of optimizing invariance-inducing objectives when the protected group label (analogous to our nuisance variables) is noisy. Given bounds on the level of label noise, this work proposes optimizing an objective based on the *distri-*

---

3. The MIMIC critical-care database is available on Physionet (Goldberger et al., 2000).

*butionally robust optimization* framework (Namkoong and Duchi, 2016). Additionally, if given a small amount of true labels the authors suggest fitting a model to de-noise the noisy group labels and re-weight examples in the objective, which is similar in spirit to our work. In our approach, however, we exploit structural assumptions about the missingness process to build a doubly-robust estimator of the MMD penalty used during optimization.

Lahoti et al. (2020) optimize worst-case-over-groups performance without known group labels. They rely on the assumption that groups are *computationally-identifiable* (i.e. that there exists some function on the data that labels their protected group membership) (Hébert-Johnson et al., 2018) and use a model to identify groups on which performance is worst. They pose an adversarial optimization between the group-labeling model — which searches for groups with poor performance — and the primary predictive model. Inspired by this work, Creager et al. (2021) find worst-case group assignments based on an empirical risk minimization (ERM) model that maximizes invariance penalties and Ahmed et al. (2020) illustrate that this objective performs well on a wide range of benchmarks. Relatedly, Liu et al. (2021) run usual ERM training and then a second iteration of ERM that upweights the loss for datapoints on which the model performs badly. This identifies groups with bad model performance without explicit group labels. In both of these works, the groups could be seen either as a nuisance variable or as a confounder that correlates the label and some nuisance variable. However, in our setting (and in that of Makar et al. (2021); Veitch et al. (2021); Puli et al. (2021)), in exchange for being willing to make assumptions on the test distribution family, we do not need to observe samples with poor model performance at training time (and may not see any) to prevent sudden decreases in performance on held-out data at test time.

## 6. Conclusion

We present estimators for the MMD that extend recent invariant prediction methods to missing data. Unlike prior estimators that only leverage data with nuisances observed, or consider worst-case estimation, the presented estimators of the full data objective are consistent when either auxiliary model can be learned. As we show in proposition 2, estimation of the full data objective is necessary to preserve the theoretical properties of invariant prediction methods. In the experiments, the DR and REG estimators are able to match full-data MMD performance and improve test accuracy relative to the OBS estimator. In practice, we recommend exploring the two simpler proposed estimators (REG and IP) in addition to the DR estimator and selecting the model based on the validation metric.

Moving forward, one limitation is that the full-data estimator — used as ground-truth MMD evaluation for the experiments — may itself have high variance on small datasets with strong nuisance-label correlation. Variance reduction is an important avenue both for optimizing and evaluating with the MMD using smaller batch sizes (in our experiments, batch sizes 1500 for MNIST and 4000 for MIMIC are large). Beyond variance reduction, it is a promising direction to apply the methodology in this work to the mutual information objective in Puli et al. (2021), which sidesteps the choice of kernel and may be better suited for continuous and high dimensional nuisances.

## Acknowledgments

The authors thank Scotty Fleming, Joe Futoma, Leon Gatys, Sean Jewell, Tayor Killian, and Guillermo Sapiro for feedback and discussions. This work was in part supported by NIH/NHLBI Award R01HL148248, NSF Award 1922658 NRT-HDR: FUTURE Foundations, Translation, and Responsibility for Data Science, and NSF Award 1815633 SHF.

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
