# OpenReview forum: "Learning Invariant Representations with Missing Data"
_cclear.cc/CLeaR/2022/Conference — CLeaR 2022 Poster_

### Official Review · Reviewer_mmQ5 · 2021-11-10

**Confidence:** 3
**Overall Score:** 6

**Main Review:**

The paper proposes a simple modification to the loss function of [Veitch et al, 2021] when learning models invariant to spurious features, to handle missing spurious features in the training data. The paper is clearly written and easy to follow. This review focuses only on its weaknesses.

- All experiments are run with lambda = 1 (trade-off between MMD objective and class-prediction objective), but in any practical problem practitioners will tune lambda. For some missingness mechanisms (e.g. Z is revealed with constant uniform probability p), missing-MMD and regular MMD (or MMD on observed data alone) may induce the same ordering on h_X, but have different magnitudes -- lambda = 1 may not be well-tuned for one of them, and may explain poorer downstream prediction performance.
- Prior work makes an identifiability assumption (i.e., Z is computable via some fixed function of (X,Y) [Lahoti et al, 2020]). The paper instead makes an "instrument"-type assumption: the only effect of Z on it's missingness \Delta is through the observable X. Figure 2a, 2b show that identifiability can be violated while still satisfying the "instrument"-type assumption. However, it is not clear whether these assumptions hold in practical datasets. For Experiments 4.2, 4.3, the paper should benchmark eventual prediction quality against the distributionally robust baselines (e.g. some of the works following [Lahoti et al, 2020]). The variance experiment of Sec 4.4 will be particularly informative to understand the trade-off between (1) fixing missing Z by re-weighting with G as in the paper; (2) imputing potentially noisy Z as done by Wang et al 2020; (3) using training samples with poor model fit to drive iterative ERM procedures as in Liu et al 2021.


**Summary:**

Simple modifications to loss functions for learning invariant models to deal with missing data

---

> ### Author Response · Authors · 2021-12-04
> **Response to Reviewer mmQ5**
>
> **[clearly written and easy to follow]**
>
> We are glad you think so!
>
> **[all experiments run with lambda=1, but scale can vary depending on missingness mechanism]**
>
> In the appendix of the submission, we also included lambda=5. However, we agree that this should be explored further as in practice one needs to tune lambda. We are adding a simulation in the appendix that studies the effect of changing lambda for a given method/dataset. Thanks for the suggestion.
>
> RE magnitude of observed-only MMD vs missing-MMD when missingness is uniformly random, these estimates should be the same in magnitude on average due to both estimators being unbiased. But we may have misunderstood this comment. Is there a different source of magnitude difference that you were thinking of?
>
> For your next two questions, addressing them is crucial for improving the clarity of our work. Please allow us to respond in detail.

---

> > ### Author Response · Authors · 2021-12-04
> > **Continued Response**
> >
> > **[missingness assumptions, do they hold in practice?]**
> >
> > The ignorability assumption we make — Z indep Delta | other variables X,Y — goes by several names in the literature and is the same as “no unobserved confounders” in causal effect estimation and is an instance of the general “missing at random” or “coarsening at random” [Rubin/Little or Tsiatis textbooks]
> >
> > This assumption is usually necessary for identification/estimation of functionals involving Z (MMD in this case) unless one is willing to make very strict assumptions about the data generating process (e.g. that the Z|X,Y conditional takes so and so exact parametric form).
> >
> > Still— as your review correctly suggests, identification should not be used to justify a claim that a particular assumption holds of the data distribution.  Instead, practitioners should reason with domain experts to discuss plausibility. Allow us to consider the MIMIC experiment in the text with Z indicating race.
> >
> > Ignorability would be satisfied, by definition, if the propensity not to report race is due only to non-race variables in X,Y (e.g. other demographics or health statuses). The example in our experiment is that male-identified people report race less. Ignorability would not be satisfied for example if certain racial groups reported race less in a way that is not explained fully by non-race variables in X. This implies that ignorability is more likely to be satisfied as one collects more features in X that could explain the missingness (e.g. socioeconomic status).
> >
> > There is therefore a dilemma for satisfying ignorability. We do expect *some* non-race socially-relevant variables to explain missingness of race reporting, but it may be hard to find these variables. Increasing the X feature set may satisfy ignorability but with several trade-offs [D’Amour 2021] including violating positivity, and in the case of learning the m(X) and G(X) models in our estimator, higher-dimensional regressions.
> >
> > In summary, one first chooses Z that they want to apply this methodology to (because they think it may cause generalization problems in a new test distribution). Next, one should ask if it is possible to enforce the objective. If Z is subject to missingness, answering this question involves explicitly picking the feature set X so that ignorability may plausibly be satisfied. This in turn may necessitate talking to a specialist about Z and its missingness. We note that if the answer to this question is no, this seems also to indicate that Z cannot only be a function of X and Y, violating the assumptions in the other papers mentioned.
> >
> > **Your comment has rightly prompted us to add discussion of this process to the text. Thanks!**
> >
> > [D’Amour 2021 Deconfounding Scores: Feature Representations for Causal Effect Estimation with Weak Overlap]
> >
> > **[should benchmark against DRO-with-missingness baselines, for example those that impute Z, Wang et al 2020]**
> >
> > This comparison involves two distinct questions.
> >
> > First, the underlying generalization methods (our/Veitch’s independence condition vs. group-DRO objectives) promise generalization on different (sometimes overlapping) sets of test distributions. This is true even without any missingness.
> >
> > Next, our missingness method and those in recent group DRO papers make different assumptions about missingness.
> >
> > In this work, rather than propose the generalization method, we focus on missingness methodology. To compare missingness work rather than underlying generalization methods, we instead propose the following: Estimate the recommended Group DRO objectives using our missingness assumptions to show that our missingness methodology is not specific to the MMD objective. Then, we can compare Group DRO objectives estimated in two ways under missingness (our estimators versus e.g imputation in Wang et al 2020 as suggested by the reviewer).
> >
> > **What do we expect:** as mentioned by the reviewer, we propose a re-weighting estimator. However, we also propose the Regression and DR estimators, which also make use of learning E[Z|X] where Z is subject to missingness.  This means our proposed regression and DR estimators are expected to work when learning E[Z|X] is possible, which is also when imputation methods are expected to work. The only difference is we require ignorability rather than Z being a deterministic function of X,Y as required by the papers mentioned by the reviewer.

---

### Official Review · Reviewer_YW7q · 2021-11-22

**Confidence:** 4
**Overall Score:** 6

**Main Review:**

Summary:
This paper proposes missing-MMDs, a new objective for invariant prediction methods that measures nuisance model dependence under missingness. Authors show the consequences of naively enforcing independence on only the nuisance-observed subset of data; derive the missing-MMD and show that it is consistent when one can predict either the missingness variable or the nuisance; show that missing-MMDs improve test accuracy over using observed-only data and perform close to ground-truth estimation with no missingness.

Main Review:
The main contribution is to present missing-MMDs, an optimization objective that extends the scope of recent invariant representation learning methods. In practice, practitioners are recommended exploring the two simpler proposed estimators in addition to the DR estimator and selecting the model based on the validation metric.

Strength:
1)This is an overall well-written paper, with good novelty and clearly described preliminaries.
2)The proposed method is effective as demonstrated by the experimental results.
3)The motivation of the proposal of missing-MMDs is reasonable. To improve generalization on a range of test distributions, it is necessary to handle missingness appropriately.
4)The analysis of extending these invariant prediction methods to handle missing data is not straightforward is thorough and insightful. This difficulty stems from the invariant method’s optimization objective, which usually includes a measures of dependence.

Concerns:
1)Too few experiments. Experimental results on more datasets are needed to present better.
2)Tables need described captions.
3)Minor question:‘0.00 ± 0.01’ in Table 2, is the minimum one (from three seeds) is negative?

**Summary:**

This paper proposes missing-MMDs, a new objective for invariant prediction methods that measures nuisance model dependence under missingness. Authors show the consequences of naively enforcing independence on only the nuisance-observed subset of data; derive the missing-MMD and show that it is consistent when one can predict either the missingness variable or the nuisance; show that missing-MMDs improve test accuracy over using observed-only data and perform close to ground-truth estimation with no missingness.

---

> ### Author Response · Authors · 2021-12-04
> **Response to Reviewer YW7q**
>
> **[The analysis of extending these invariant prediction methods to handle missing data is not straightforward is thorough and insightful]**
>
> Glad to hear the analysis built some insight !
>
> **[Too few experiments: more datasets]**
>
> Thank you for this feedback. Do you have any particular suggestion on a kind of dataset to focus on for an additional experiment? As mentioned to reviewer UaPh, we will add an experiment with real, rather than simulated missingness. This could be done either in MIMIC with a different choice of Z, or another dataset. Would be glad about your further input on this in order to round out the paper.
>
> **[Table captions]**
>
> We have updated the table captions with more information. Thanks for catching this.
>
> **[Minor question:‘0.00 ± 0.01’ in Table 2, is the minimum one (from three seeds) is negative?]**
>
> Great question! Yes. Technically, the estimators we propose are unbiased (when the auxiliary models are correct). This means that when the true MMD is 0 or close to it, it is definitely possible that a given estimate can be negative even though the true MMD is non-negative, though this should become less of an issue as sample size increases due to consistency. [Gretton et al. A Kernel Two-Sample Test] discuss precisely this question and give a few alternative biased estimators that are non-negative. We would be glad to add these into the experiments if it seems useful, as the missingness methodology can also be applied to them under the same assumptions.

---

### Official Review · Reviewer_UaPh · 2021-11-26

**Confidence:** 3
**Overall Score:** 6

**Main Review:**

**Summary**
This submission proposes a training objective for learning invariant representations under spurious correlated data settings, especially when the nuisance data is missing.

**Main review**
The methodology described in this work is interesting, with solid theoretical support and promising empirical performance which is consistent with their claim.
The originality is not that much, as similar ideas (punishing some distance between different domain / demographics) have long been explored in the literature. Some examples include [1] [2].
The problem of missing data is reasonable, and could relates to the fundamental problem of causal inference.
The derivation of the MMD estimators is correct.
The numerical results are somewhat toy, but still reasonable.
PS. Some references in related topics, such as [3] and [4], are missing, though.

[1] Yaroslav Ganin, Evgeniya Ustinova, Hana Ajakan, Pascal Germain, Hugo Larochelle, François Laviolette, Mario Marchand, Victor Lempitsky. Domain-Adversarial Training of Neural Networks.
[2] Victor Veitch, Alexander D’Amour, Steve Yadlowsky, and Jacob Eisenstein. Counterfactual invariance to spurious correlations: Why and how to pass stress tests
[3] Zhang, Dinghuai, Kartik Ahuja, Yilun Xu, Yisen Wang and Aaron C. Courville. “Can Subnetwork Structure be the Key to Out-of-Distribution Generalization?” ICML (2021).
[4] Ahuja, Kartik, Ethan Caballero, Dinghuai Zhang, Yoshua Bengio, Ioannis Mitliagkas and Irina Rish. “Invariance Principle Meets Information Bottleneck for Out-of-Distribution Generalization.” ArXiv abs/2106.06607 (2021)

**Significance**
The problem of out-of-distribution generalization and spurious correlation is an important one. This topic (together with doubly robust estimators, etc) is closely related to CleaR community, as causality lies in the center of OOD generalization. The methodology would potentially have impacts outside CLeaR community.

**Clarity**
The writing is clear but some sentences can be improved if they are not too complicated (scientific writing should have simple structure).


**Summary:**

Review

---

> ### Author Response · Authors · 2021-12-04
> **Response to Reviewer UaPH**
>
> Thanks for the positive comments! We focus on responding to your questions and concerns.
>
> **[originality not much, punishing distance between domains / demographics explored in e.g. [1] , [2] ]**
>
> We agree this has been explored. In this work, we do not aim to present the technique of enforcing independence to domains/demographics as our novel idea. Instead, we aim to extend existing works that do so. [2, Veitch et al] that you mention is our main starting point and we extend their objective to the case where Z is subject to missingness. We have revised the draft to add clarity that we are not introducing the independence constraints ourselves. We have also added discussion of [1, Ganin et al]: namely, it is one of the starting points for works like [2] that we build on.
>
> **[numerical results are somewhat toy, but still reasonable]**
>
> We would be glad to incorporate any detailed feedback you have on this point. Any suggestions on what in particular gave this impression and how it could be improved?
>
> One direction is to use data with real, rather than simulated, missingness. In this case, we would have to give up the ability to report ground truth full-data MMD estimates and instead focus only on reporting test accuracy.
>
> **[some references in related topics, such as [3] and [4], are missing]**
>
> Thanks for the pointer to these relevant papers! We have incorporated them into the draft. In particular, [4. Ahuja et al] operate in a similar setting as us, relative to papers like [Arjovsky, Invariant Risk Minimization], in that Ahuja and we both make particular assumptions about the data generating process and enforce suitable independencies that entail generalization on certain distributions we may anticipate at test time.

---

### Author Response · Authors · 2021-12-04
**Summary of Author Rebuttal to Reviews**

Dear Reviewers and Meta-reviewer. We thank the reviewers for their detailed comments. We summarize the positives and concerns from the three reviews here. We then summarize our proposed agenda for rounding out the paper.

**[Positives]**

**Reviewer UaPH**
- interesting, with solid theoretical support and promising empirical performance
- “closely related to CleaR community” and “potentially have impacts outside CLeaR”

**Reviewer YW7q**
- well-written, with good novelty and clearly described preliminaries
- effective as demonstrated by the experimental results.
- analysis is thorough and insightful.

**Reviewer mmQ5**
- clearly written and easy to follow

**[Critique]**

- UaPH says the originality is not that much because enforcing independence to demographics has been explored. We agree and do not make the claim to introduce this paradigm. Instead, we take existing work as a starting point. In particular, the reviewer suggests [2] which is the main work we build on and we discuss it throughout our text.
- UaPH and YW7q ask for more datasets, we have responded by asking for more details on which kind of experiment seems to be missing.
- mmQ5 asks for more hyper-parameters, some of which are already included in the appendix and we have begun including more in the appendix.
- mmQ5 asks to elaborate on when the ignorability assumption is expected to hold, which prompted us to describe this in more detail and has improved the text.
- mmQ5 asks to benchmark against group DRO methods that also handle missingness. See our response to the reviewer for our proposed comparison

**[Summary of proposed experiments]**
- Add one more dataset. This one can have real missingness. This means we give up the ability to report ground truth full-data MMD and instead report only accuracy.
- Adapt one of the biased non-negative MMD estimators from Gretton et al and apply missingness methodology to it, as an alternative to our unbiased estimators
- Adapt one Group-DRO method with our missingness methodology
- More hyperparameters for Lamba on simulations in the appendix.

---

### Decision · Program_Chairs · 2022-01-13

**Decision:**

Accept (Poster)

**Comment:**

Following existing work (by Veitch et al), the paper considers the problem of learning, from observed data (X_1, Y_1, Z_1, ..., X_n, Y_n, Z_n), a predictor f(X) for a variable Y, while enforcing, at least to a certain degree, that f(X) \independent Z | Y. The paper's main contribution is in dealing with a situation, where some of the Z_i's are missing, by modifying the independence measure (here MMD) accordingly. It seems that the authors have not provided code during submission time. I regard the topic of the paper interesting. The reviewers mentioned many important points and the scores resulted in a borderline case, with all reviewers leaning towards acceptance. Personally, at several places, I find the presentation of the paper not sufficiently clear and I hope that the authors put in some effort to improve on that aspect (some points are highlighted below but there are more in the paper). Also, I have asked another expert for their opinion on the paper. They mentioned the following two points: (1) "It's not obvious to me whether we should think of missigness as caused by X or a cause of X. (In the mimic example, missingness is basically just a part of X). In the latter case, an obvious thing to do is just augment Z to be \tilde{Z}=(Z, missingness) and then enforce the independence with respect to \tilde{Z}. I'd have liked to see this run as a baseline." and (2) "The experiments seem a bit weak. In particular, the results from these MMD regularizations are pretty sensitive to regularization strength and (to a lesser degree) kernel choice. It seems important to vary these to show that the results hold up." I recommend that the authors address these two points, together with the other points mentioned by the reviewers.


Additional points on presentation.
- The document should be carefully checked for language mistakes and typos (including usage of parentheses and punctuation).
- Fig 1: I find this figure unclear, please clarify all terms and rewrite the caption.
- For most of the presentation, the nuisance is assumed binary (apart from App G, which I found a bit short). It should be mentioned in the abstract that the main focus lies on binary nuisance variables.
- In my opinion, the term "depends on" should refer to statistical dependence.
- Fig 2 shows graphs -- in which sense are these "generative processes"? Also, please comment on the directed cycle in Fig 2b.
- "measures of dependence — e.g., Maximum Mean Discrepancy (MMD)" -- please clarify (in general, MMD is a distance measure between distributions) and specify the relation to HSIC.
- "We work within the framework of Veitch et al. (2021) because their method applies to a more general version of the model in Makar et al. (2021)." -- please clarify.
- "The presented method can extend to (...) other graphs (e.g. Figure 2(b))" -- please clarify.
- "For exposition, let ... be distributed given just X rather than (X, Y)" -- please clarify.
- "(conditioning on Delta = 1 okay by ignorability)" -- please clarify.
- Please ensure that the statements are mathematical in that all terms are either defined or related to a quantifier (this concerns, e.g., almost all propositions).
- "for each X" -- please clarify.
- "This is particularly bad when" -- please clarify.
- The notation using MMD is inconsistent (IMO, the notation MMD(X1, X2) is incorrect). Also, the subscript changes regularly.
- X is used for random variables and for real numbers.
- "E(Z = 1|X)" -- please fix.
- The usage of the term "estimator" is at least confusing (see, e.g., App F).
- IMO, the presentation of the proofs should be improved, see, e.g., F.1. This includes notation and the usage of assumptions, such as Delta independent Z, given X,Y.
- Please be careful to distinguish between random variables and random numbers and between probability distributions and densities.
- The references are not up-to-date.